# Husk-like Zinc Oxide Nanoparticles Induce Apoptosis through ROS Generation in Epidermoid Carcinoma Cells: Effect of Incubation Period on Sol-Gel Synthesis and Anti-Cancerous Properties

**DOI:** 10.3390/biomedicines11020320

**Published:** 2023-01-23

**Authors:** Wardah A. Alhoqail, Abdulaziz S. Alothaim, Mohd Suhail, Danish Iqbal, Mehnaz Kamal, Majid Mohammed Asmari, Azfar Jamal

**Affiliations:** 1Department of Biology, College of Education, Majmaah University, Al-Majmaah 11952, Saudi Arabia; 2Department of Biology, College of Science in Zulfi, Majmaah University, Al-Majmaah 11952, Saudi Arabia; 3King Fahd Medical Research Center, King Abdulaziz University, Jeddah 21589, Saudi Arabia; 4Department of Medical Laboratory Sciences, Faculty of Applied Medical Sciences, King Abdulaziz University, Jeddah 21589, Saudi Arabia; 5Department of Health Information Management, College of Applied Medical Sciences, Buraydah Private Colleges, Buraydah 51418, Saudi Arabia; 6Department of Pharmaceutical Chemistry, College of Pharmacy, Prince Sattam Bin Abdulaziz University, Al-Kharj 11942, Saudi Arabia; 7Research Center, College of Pharmacy, King Saud University, Riyadh 11451, Saudi Arabia; 8Health and Basic Science Research Centre, Majmaah University, Al-Majmaah 11952, Saudi Arabia

**Keywords:** sol-gel synthesis, zinc oxide nanoparticles, human epidermoid carcinoma, MTT assay, X-ray diffraction, ROS generation

## Abstract

This study effectively reports the influence of experimental incubation period on the sol-gel production of husk-like zinc oxide nanoparticles (ZNPs) and their anti-cancerous abilities. The surface morphology of ZNPs was studied with the help of SEM. With the use of TEM, the diameter range of the ZNPs was estimated to be ~86 and ~231 nm for ZNP^A^ and ZNP^B^, prepared by incubating zinc oxide for 2 and 10 weeks, respectively. The X-ray diffraction (XRD) investigation showed that ZNPs had a pure wurtzite crystal structure. On prolonging the experimental incubation, a relative drop in aspect ratio was observed, displaying a distinct blue-shift in the UV-visible spectrum. Furthermore, RBC lysis assay results concluded that ZNP^A^ and ZNP^B^ both demonstrated innoxious nature. As indicated by MTT assay, reactive oxygen species (ROS) release, and chromatin condensation investigations against the human epidermoid carcinoma (HEC) A431 cells, ZNP^B^ demonstrated viable relevance to chemotherapy. Compared to ZNP^B^, ZNP^A^ had a slightly lower IC_50_ against A431 cells due to its small size. This study conclusively describes a simple, affordable method to produce ZNP nano-formulations that display significant cytotoxicity against the skin cancer cell line A431, suggesting that ZNPs may be useful in the treatment of cancer.

## 1. Introduction

Skin serves as the protective barrier against pathogens, trauma, high temperatures, and radiation. Nonetheless, the global incidence of skin cancer is intensifying as a result of continuous exposure to ionizing radiations and toxins [1]. According to World Health Organization (WHO), the predicted global incidence of skin cancer will double to about 37 million new cancer cases by 2040 [2]. The most common form of skin cancer includes basal cell carcinoma (BCC), squamous cell carcinoma (SCC), melanoma, and Merkel cell carcinoma (MCC). Epidermoid carcinoma, the second most prevalent type of metastatic carcinoma, displays a serious, aggressive proliferation habit and a stronger propensity to spread [3]. As the abnormal growth of epidermal cells has been linked to skin cancer [4], the human epidermoid skin cancer cells (A431) have evolved as an effective targeting cell line to evaluate the anti-cancer capabilities of various formulations.

Zinc is an important mineral that functions as a cofactor in several enzymes [5]. The oxide of zinc (ZnO) belongs to group II-VI semiconducting material, present naturally as zincites [6]. It is relatively safe, inexpensive, cosmopolitan in distribution, stable at high temperature/pressure, and is very effective in providing protection against UV rays [7]. ZnO displays a direct band width of 3.37 eV, high exciton binding energies of 60 meV at 298K [8], high transmittance, good electrical conductivity, and excellent chemical and thermal stability [9]. The nano form of ZnO is readily absorbed by mammalian cells and tissues due to its large surface-to-volume ratio [10].

Zinc oxide nanoparticles (ZNPs) have gained tremendous attention in the last two decades. It is the most widely accepted nanomaterial due to its distinctivephysico-chemical and biological attributes, such as biocompatibility, eco-friendly nature, low cost, and non-toxic behaviour [11,12,13]. The United States Food and Drug Administration (FDA) has put zinc oxide on the list of safe materials when used in accordance with good manufacturing practice [14]. In recent years, ZNPs have been applied in a variety of applications including ceramics, plastics [15], paints [16], cosmetics [13], optics [17], electronics [18], photonic material [19], sensors [20], acoustic wave devices [17], and the textile industry [21]. The fabrication of ZNPs has been achieved using different techniques, including hydrothermal [22], sol-gel chemical [23], combustion [24], the spray pyrolysis method [25], microwave-facilitated processes [26], the pulsed-laser ablation method [27], as well as biological synthesis [28]. These divergent techniques are often used to create ZNPs, leading to a variety of architectural forms of nanomaterials.

The sol-gel method, a bottom-up approach, involves mixing the reacting species over a magnetic stirrer. The reaction is incubated for a stipulated time until a white precipitate appears. This precipitate is then removed by centrifugation, washed, and kept for drying. This is a straightforward and uncomplicated approach that enables control over ZNP synthesis through meticulous tweaking of process variables throughout fabrication [29,30]. This simple, well-controlled, classical approach has been used for the cost-effective production of ZNPs at room temperature [31,32,33].

Previously, the sol-gel-derived synthesis of ZNPs was accomplished by adjusting incubation temperature [9,34], stirring speed [35], as well as the introduction time of the ion carrier [36], generating ZNP architectures resembling thorns [35], flowers [34], microtubules [36], as well as spheres [9]. To the best of our knowledge, the outcome of different incubation time periods in the synthesis of ZNPs has not been studied. Accordingly, we explore the impact of incubation time on the sol-gel formation of ZNPs and their anti-cancerous properties. We have verified the fabrication of ZNPs using scanning electron microscopy (SEM), X-ray diffraction (XRD), and spectroscopic investigations. We selected human epidermoid carcinoma (HEC) cells to examine the effect of ZNPs because these skin epidermal cells are the first to be exposed to external substances, including nanoparticles [37]. Thus, we evaluated the cytotoxic activities of ZNPs prepared with varying incubation time against the model skin carcinoma cell line A431.

## 2. Materials and Methods

### 2.1. Synthesis of Husk-like Zinc Oxide Nanoparticles (ZNPs)

The experiment was carried out in a simple way to create ZNPs, utilizing DDW in two distinct reactant conditions. Synthesis A involved the mixing of 10 mL of 0.1 M solution of zinc acetate dihydrate (ZAD) (Merck, Bangalore, India), which was poured in 180 mL of DDW under a regulated stirring condition. This was followed by the addition of 0.001 moles of capping agent, cetyltrimethyl ammonium bromide (CTAB) (Merck, Bangalore, India), to the reaction mix, which was kept under stirring. Finally, 0.5 g of ionic carrier, i.e., sodium hydroxide (NaOH) (Merck, Bangalore, India), was introduced gradually into the cocktail. The setup was kept for less than an hour, until a white cloudy appearance occurred. The solution was then shifted to the centrifugation tubes. This was further kept at ambient temperature under a well-observed incubation condition for 2 weeks. Next, the product was washed with the DDW and absolute alcohol. Lastly, the sample was labelled as ZNP^A^ and stored for further characterization. In Synthesis B, all the variables remained same except for the incubation condition, which was increased to 10 weeks (at ambient temperature). Finally, the product was washed with the DDW and absolute alcohol. The product was labelled as ZNP^B^ and stored until further use.

### 2.2. Characterization of ZNPs

The X-ray diffraction pattern of produced ZNPs was examined using a Bruker D8 ADVANCE X-ray diffractometer (Karlsruhe, Germany) with an X-ray (Cu-K_α_ radiation) wavelength (λ) of 1.54178, a step dimension of 0.01°, and a scanning speed of 0.02 steps/s. The power-generating parameters were tuned to 40 kV and 40 mA. The absorption spectrum was obtained utilizing a dual-beam LAMBDA 365 UV/Vis spectrophotometer (Perkin-Elmer, Houston, TX, USA). DD water was used as a reference for the background correction. Scanning electron microscopy, JEOL JSM-6510 LV (Tokyo, Japan), and transmission electron microscopy, JEOL JEM-2100F (Tokyo, Japan), were used to evaluate the size and morphology of the nanomaterials. The zeta potential of the ZNPs was determined using a Zetasizer Nano ZS instrument (Malvern, Hampshire, UK).

### 2.3. Red Blood Cells (RBCs) Lysis Assay

A known number of RBCs (~2 × 10^8^ cells/mL) were mixed with 1 mL of ZNPs at various concentrations in a final volume of 2 mL and incubated at 37 °C for 2 h in triplicates. After a stipulated time period, the reaction mixture was centrifuged at 1200 g, and the supernatant was obtained. To check for hemoglobin leakage, the absorbance was examined at 576 nm. The extent of lysis obtained with Triton X-100 (0.1%) was taken as 100%. The percentage of the following equation was used to report the RBC lysis result [38]:$$\%\;\mathrm{RBC}\;\mathrm{lysis}=\left(\frac{{\mathrm{Abs}}_{T}-{\mathrm{Abs}}_{C}}{{\mathrm{Abs}}_{100\%}-{\mathrm{Abs}}_{C}}\right)\times 100$$
where, Abs_T_ is the absorbance obtained from ZNPs, Abs_C_ is the absorbance of the control (PBS), Abs_100%_ is the absorbance in the presence of 0.1% Triton X-100.

### 2.4. Cell Lines and Culture

The cell lines, human epidermoid carcinoma A431 and kidney epithelial Vero, were obtained from the National Center for Cell Sciences (NCCS), Pune, India and grown in Dulbecco’s Modified Eagle Medium (DMEM)-F12 medium, supplemented with 10% (*v*/*v*) fetal calf serum (FCS), sodium bicarbonate (NaHCO_3_) (1.5 g/L), and L-glutamine (2 mM).

### 2.5. Cell Viability Assay

The MTT [(3-(4,5-dimethylthiazol-2-yl)-2,5-diphenyltetrazolium bromide) tetrazolium] (Sigma, Bangalore, India), assay was utilized as an indicator of cytotoxicity of ZNPs (ZNP^A^ & ZNP^B^), as reported earlier [36]. An overnight culture of 1 × 10^4^ cells seeded in a 96-well culture plate containing 100 µL of DMEM-F12 was treated with 5, 10, and 25 µM of various ZNP doses in triplicates and incubated for 24 h. Cells incubated in the presence of media were taken as control. After incubation, 10 μL MTT was introduced to cells from a stock solution (5 mg/mL dissolved in PBS, pH 7.4) and colour development was examined. Finally, after adding 100 μL DMSO (Merck, Bangalore, India), to every sample in order to dissolve the blue precipitate, the percentage of viable cells was calculated as previously mentioned [39]:
$$\%\;\mathrm{Cell}\;\mathrm{viability}=\left(\frac{{\mathrm{OD}}_{\mathrm{control}}-{\mathrm{OD}}_{\mathrm{treated}}}{{\mathrm{OD}}_{\mathrm{control}}}\right)\times 100$$


### 2.6. Reactive Oxygen Species (ROS) Generation

The ROS value was determined as per an earlier report [40]. A known number of cells (1 × 10^4^/well) were tested against diverse ZNPs for 12 h. After a stipulated time period, the cells were mixed with 10 μM of Dichloro-dihydro-fluorescein diacetate (DCFH-DA) (Merck, Bangalore, India), for 30 min in the dark. Images of the intracellular fluorescence intensity were captured using an inverted fluorescence microscope (Nikon ECLIPSE Ti-S, Tokyo, Japan). For quantitative analysis, cells were cultured for 12 h in a 96-well black-bottom culture plate in the presence of various ZNPs. After staining with DCFH-DA for 30 min and washing with PBS (200 µL, pH 7.4) to dispose off excess stain, the florescence intensity was finally measured using FLUOstar^®^ Omega microplate reader (BMG Labtech, Ortenberg, Germany). The excitation and emission wavelengths were set at 485 and 528 nm, respectively. The fluorescence intensity of treated and control cells was determined with ImageJ software (NIH, Bethesda, MD, USA) using the following formula:
$$\mathrm{Fluorescence}\;\mathrm{intensity}\;\left(\%\right)=\left(\frac{\mathrm{Fluorescence}\;\mathrm{intensity}\;\mathrm{of}\;\mathrm{treated}\;\mathrm{cells}}{\mathrm{Fluorescence}\;\mathrm{intensity}\;\mathrm{of}\;\mathrm{untreated}\;\mathrm{cells}}\right)\times 100$$

### 2.7. Fluorescent Nuclear Staining

The cytotoxic impact of ZNPs in inducing apoptosis in carcinoma cells was assessed using 4′,6-diamidino-2-phenylindole (DAPI) dye by incubating them with various ZNPs [41]. After 24 h incubation, the cells were fixated for 10 min with 4% paraformaldehyde and resuspended with a permeabilizing solution comprising 0.5% Triton X-100 reagent and 3% paraformaldehyde. An inverted fluorescence microscope was then employed to image and quantify apoptotic cells (Nikon ECLIPSE Ti-S, Tokyo, Japan).

### 2.8. Statistical Analysis

The data are shown as the mean ± SD of three separate tests (*n* = 3). Graph Pad Prism (V 6.01) was used to evaluate the significance of the findings using one-way ANOVA and Dunnett’s Multiple Comparison Test. A *p*-value of ≤0.05 was regarded as significant.

## 3. Results and Discussion

ZNPs represent a class of NPs extensively employed as candidates in cancer detection and chemotherapy because of their peculiar physical and chemical characteristics. There are several techniques, e.g., chemical, physical, or biological methods that can be utilized to synthesize ZNPs. Physical processes involve vapor transport [42], arc plasma [43], pulsed-laser deposition [44], and ultrasonic irradiation [45]. Co-precipitation [46], microemulsion [47], sonochemical method [48], sol-gel process [35], chemical vapor deposition (CVD) [49], and hydrothermal procedures [50] are a few examples of chemical-based methods [51]. An alternate procedure for obtaining ZNPs involves green synthesis using biological samples [52]. The confluence of such varied fabrication methods yields a collection of distinct nanostructured materials. Nevertheless, majority of such procedures require a significant amount of expertise as well as specialized equipment, which enhances the cost. Some of the studies involved a high incubation temperature of 120 °C for 24 h [53], an annealing temperature of 450 °C for 8 h [54] or 800 °C for 60 min [55], as well as heating at 500 °C for 2 h in a furnace [9]; these methods hinder widespread usage due to complicated procedures. Near-room-temperature synthesis is cost-effective, avoiding complex apparatuses [56] with no compromise on the activity [34,57].

The conventional synthesis of low-cost and high-surface-area metal oxide nanomaterials is normally performed either by precipitation or sol-gel synthesis. The sol-gel method relies on hydrolysis and condensation processes using metallic alkoxides M(OR)*_n_* as a source of ions [58,59]. The sol-gel method allows the mixture of the initial reagent on an atomic level; this reaction generates an infinite, dense, 3D crystalline structure [60]. This methodology is simple, economically friendly, easily tunable, requires ambient temperatures, shows good reproducibility, reduces the possibility of impurities, and is a facile way to generate complex nanocomposites and nanostructures [33,61,62]. The resulting nanostructures can be obtained as particulate powdered particles, fibers, or thin films/sheets [62]. Zinc belongs to a class of elements that readily forms polymeric hydroxides, which is a fundamental requirement for the sol-gel process [59]. As a result, this method is widely used by scientists to manufacture ZnO-based nanoparticles [63].

Broadly speaking, nanoparticle formation is a complicated procedure with several variables involved that could impact their properties. As such, the action of ZNPs is dependent on the particle size, morphology, pH, and biocompatibility; while its size and morphology can be effectively controlled by the pH of the reaction mixture, the type of solvent used, the reaction temperature and time, as well as the ratio of reacting species [59,64]. Previous research has shown that the variation in the introduction time of the ion carrier (5 min and 10 min) [36] as well as stirring speed (e.g., 500 rpm, 1000 rpm, 1500 rpm, and 2000 rpm) [35] led to the fabrication of microtubule-like [36] and thorn-like [35] patterns, respectively, via the sol-gel method at room temperature. Prompted by these investigations, we have evaluated the effect of varying the incubation period on the synthesis and the performance of ZNPs. Our simple design does not require expensive materials or sophisticated machineries. Moreover, the effects of altered incubation periods on the anti-cancer property of ZNPs against skin cancer have not been previously evaluated.

### 3.1. Characterization of ZNPs

#### 3.1.1. UV-Visible Spectroscopy Analysis of ZNPs

UV-visible spectroscopy is an effective method for investigating the opto-structural features of ZNPs at room temperature. The samples were scanned at a wavelength range of 350–650 nm, and the absorption peak (λ_max_) was recorded. Metallic nanomaterials display a wide surface plasmon resonance (SPR) absorbance maximum in the ultraviolet (UV) region of the spectrum [65,66]. The ZNPs exhibit narrow absorption bands around 379 and 388 nm (UV-A region), respectively, without any additional peaks, indicating coupled light waves as well as the electron vibrations of nanoparticles [67,68]. This absorbance peak corresponds to the optical band gap of ZnO [9]. The detected peak values depict wurtzite structural characteristics [69].

Nonetheless, the intensity of the absorption bands increased from ZNP^A^ to ZNP^B^ when incubation time was increased from 2 weeks to 10 weeks (Figure 1). Interestingly, it was found that extending the incubation time during the synthesis process resulted in a slight change in the λ_max_ of ZNPs. This may be due to the increase in the diameter of the nanoparticles upon increasing the incubation time of the reaction mixture [67], leading to a change in aspect ratios, thus causing the characteristic blue-shift [70,71]. The blue-shift in the absorbance could be because of light diffraction spurred on by the crystal growth and development of grains [55]. Due to the increase in the overall dimensions of ZNP^B^, an upsurge in absorption was also observed, as incubation time is one of the factors affecting the size and form of ZNPs. It has previously been noted that the sizes and geometries of the nanoparticles affect their spectral attributes [72,73,74].

#### 3.1.2. XRD Analysis of ZNPs

The XRD analysis was employed to illustrate the crystal characteristics of ZNPs (Figure 2). The JCPDS pattern for ZnO was used as the basis for the X-ray diffraction profiles for ZNP^A^ and ZNP^B^ (File no: 043-0002). The generated ZNP nanostructures have distinct peaks, which point to a single phase. Prominent peaks produced by the ZNP structures are indicative of nanoscale dimensions [59]. The absence of any extra peaks other than those distinctive of ZNPs indicates the absence of contaminants in the specimen [57]. Furthermore, the crystalline character of the ZNPs is confirmed by the strong, constricted diffraction pattern [67].

The characteristic peaks for ZNPs at (100), (002), (101), (102), (110), (103), (200), and (112) corresponded to the 2θ values of 31.78°, 34.44°, 36.28°, 47.58°, 56.64°, 62.86°, 67.94°, and 69.02°, respectively. This leads the ZNPs to have an epitaxial state with a hexagonal wurtzite arrangement [75]. Previous research indicated that the largest peak of 2θ came at 36.3°, documented along alignment (101) [35]. Additional evidence suggesting wurtzite arrangements is provided by peaks at (002), (102), (110), and (103) [76]. This supports previous findings and demonstrates its purity [77,78]. The crystallite size was determined employing the Debye–Scherer equation, using the peak with maximum intensity (101) [79]. The ZNP^A^ and ZNP^B^ specimens are reported to be around ~21 nm and ~289 nm in average diameters, respectively. Sample thickness was found to be positively correlated with the increase in the incubation period. It was noted that prolonging the experimental incubation time produced ZNPs with increased diameters (Table 1).

The potential indicator of the stability of the colloidal solution is the magnitude of the zeta potential (−30 mV to +30 mV) [80]. Thus, the synthesized ZNPs had a considerable level of stability, as seen by the high negative zeta potential of −30.5 ± 2.5 mV and −21.0 ± 1.5 mV for ZNP^A^ and ZNP^B^, respectively. The zeta potential depicts the strength of attraction or repulsion between nearby, similarly charged particles in dispersion [81]. The stability of ZNPs results in a durable formulation, provided that they are agglomerates and not aggregates in an aqueous environment [82,83].

#### 3.1.3. SEM Analysis of ZNPs

SEM images of ZNP specimens showed the husk-like structure (Figure 3). A significant portion of strands are organized within every clump in morphologically thread-like patterns. The average thickness of these patterns of ZNP^A^ and ZNP^B^ specimens was around ~23 ± 0.5 nm and ~219 ± 0.4 nm, respectively. This is consistent with actual XRD data (Figure 2), which show that the variation in the ZNP specimens is a result of extending the incubation period in the experimental setup.

The primary ZnO nanocrystallites (~10 nm) constitute the secondary ZnO nanoparticles that can be observed in SEM pictures. The following two processes may be utilized to fuse the primary nanocrystallites and generate a larger secondary particle [84]:Fusion between two primary crystallites (~10 nm);Aggregation of the fused primary crystallites (each around ~10 nm).

A longer incubation time resulted in a more efficient and even dispersion of ions under the influence of the stabilizing agent, resulting in growth along the transverse direction (c-axis). This had an impact on the nano-diameters of ZNPs as well. Extending the incubation period of the experimental setup can result in a discrete crystallization pathway, an important determinant of overall nanocrystal dimension. Therefore, it is discovered that for ZNP specimens, a configuration resembling a husk occurs when the incubation period is increased from 2 weeks to 10 weeks [85,86].

#### 3.1.4. TEM Analysis of ZNPs

TEM data showed ZNPs arranged in a husk-like manner with average diameters of ~86 nm and ~231 nm for ZNP^A^ and ZNP^B^, respectively (Figure 4). This is in agreement with the SEM data (Figure 4) and matches well with previous research [34,35,57]. Previously, it was shown that the crystal size estimated from TEM data increases with an increase in one of the variables, e.g., annealing temperature [9]. Similarly, prolonging the incubation period of the experimental setup results in distinct morphology and size of ZnO nanostructures (Figure 4), as incubation time is one of the conditions that influences the shape and dimension of ZNPs [87,88]. Despite diverse preparation techniques reported in the past for the synthesis of ZnO nanostructures, the outcome was similar, generating flower-like structures when the same precursor (ZAD) was used [35,89,90].

A discrepancy in the particle diameter estimated by electron micrograph and the crystalline size reported by XRD was observed. This could be due to the fact that XRD simply measures the average diffraction domain size or the crystallite size, as opposed to the particle diameter calculated by SEM and TEM. While XRD usually indicates a dimension that is smaller than it should be or delivers the actual size in the absence of agglomerations (the Scherrer equation is rather a rough approximation), electron microscopy usually shows the actual size [91,92].

Zinc oxide is arranged spatially in three different crystal formations: hexagonal wurtzite, the most stable form, cubic zinc blende, and the occasionally detected rock-salt structure [93,94]. Wurtzite ZnO is a covalent crystalline structure with octahedral topology, where Zn and O atoms are alternately aligned with hexagonal orientation across the c-axis plane [59]. In addition to these crystalline structures, ZNPs may adopt distinctive morphologies (e.g., nanorings, nanocombs, or nanocages) depending on the synthesis procedure utilized [95].

### 3.2. Growth Mechanism of ZNPs

In a bottom-up approach, the sol-gel synthesis of ZNPs involves three major steps:Preparation of zinc precursor:

ZNPs are formed using Zn(CH_3_COO)_2_·2H_2_O as ion precursors. ZAD dissolved in water was slowly made to react with CTAB under a blender. This admixing generates zinc cations and acetate ions after the hydrolysis of ZAD [35].

2.Preparation of ZnO clusters:

Following the addition of NaOH, the zinc reacts with the excess of hydroxyl ions to form Zn(OH)_2_. At alkaline pH, Zn(OH)_2_ is only slightly soluble and occurs primarily as the soluble complex [Zn(OH)_4_]^2−^. These zincate ions further disintegrate into Zn^2+^ cations and OH^–^ ions [29] that are polymerized to construct “Zn-O-Zn” linkages [96]. This occurs in the abundance of OH^–^ of ion carriers that bond with the zinc ions and form ZnO [97]. The aggregation of [Zn(OH)_4_]^2−^ and CTAB together with the generation of nucleation units preceded the formation of ZNP crystals. Moreover, reaction at ambient temperature prevents rapid particle growth [10]. Subsequently, an equilibrium was reached between the hydrolysis and condensation reactions, culminating in the end product [98].

3.Crystal growth

Crystallization is a self-initiated process that occurs at ambient temperature. However, the duration of incubation strongly affects the crystal growth rate, shape, and size. When the nanoparticles/ions/crystals are saturated, the ZnO nuclei expand, creating nanostructures [99]. Due to the electrostatic force of attraction, a number of CTAB clusters with (Zn[OH]_4_)^2−^ becomes bound with ZnO edges, causing the c-axis to lengthen [50,100]. Newly generated strands gradually attach to the surfaces of the pre-existing crystalline phase, forming nano-strands as the initial structure. Due to this asymmetrical expansion, thread-like topologies develop that consolidate into husk-like ZNPs [101]. CTAB, a detergent, promotes the production of growth pockets and guides their bidirectional development as well [102].

The overall reaction mechanism is provided in Figure 1 given below [10].

CTAB binds to the different facets, e.g., [0 I Ī I], [0 Ī I 0], [Ī 0 I 0], [I 0 Ī 0], [0 I Ī 0], and [Ī I 0 0], reduces the solution’s surface tension, and slows the growth of these facets [103]. However, CTAB promotes growth along the [0 0 0 I] direction. The anisotropic nature of development results in the formation of a bunch of 2D crystal strands. The length/thickness (L/T) ratios of the husk-like crystal strands of ZNPs followed the below pattern:Length (ZNP^B^) > Length (ZNP^A^)
Thickness (ZNP^B^) > Thickness (ZNP^B^)

When incubated for two weeks, the length and thickness of ZNP^A^ are suitably controlled by the surfactant, thus promoting growth along only one axis. This leads to the growth of needle-like structures with a lower L/T ratio. As the incubation period is increased to 10 weeks, the growth of crystal is no longer under the control of the stabilizer, leading to an increase in both the length and thickness, hence giving rise to husk-like 2D crystal strands. This was due to the unsupervised growth of the facets [0 I Ī I] and [0 I Ī 0], as the capping agent now alters the binding of precursor ions along these facets by altering the free energy.

### 3.3. Anti-cancer Activity of ZNPs

A plethora of reports are now available providing evidence of ZNPs as a chemotherapeutic agent to treat cancers [104,105,106,107]. ZnO nanoparticles displayed intrinsic targeted inhibitory effects towards a plethora of cancer cells, including ovarian [108,109], lung [110], lymphoma [111], and laryngeal cancer [112], because of their distinctive chemical and physical properties. Recently, ZNPs prepared through green synthesis displayed superior inhibitory action against A431 cells when compared to Vero cells [113,114]. However, none of the previous investigations examine the influence of incubation duration on the anti-cancer capabilities of ZNPs against A341 skin carcinoma cells.

#### 3.3.1. Toxicity Evaluation of ZNPs

Prior to introducing this formulation into a medical setup, we have analyzed the concentration-dependent hemolytic effect of ZNPs (Figure 5). 

ZNP^A^ exhibited 15.68 ± 1.153 and 27.85 ± 2.30% RBC cell lysis at concentrations 50 and 100 μM, respectively. Similarly, ZNP^B^ displayed 24.95 ± 10.46 and 38.62 ± 8.07% hemolysis at the same concentrations tested. At a much higher concentration of 200 µM, significant hemolytic activity (44–75%) was observed (Figure 5a). Moreover, the data showed that ZNP^B^ has higher hemolytic activity compared to ZNP^A^ due to its large size. One study came to the same conclusion and discovered comparable results [115].

We have also investigated the apparent toxic effects of ZNPs on the normal mammalian Vero cell line (Figure 5b). A low concentration of ZNPs displayed an insignificant toxicity profile. At 50 and 100 μM, the cell viability was found to be 70.22% (*p* < 0.0001) and 66.27% (*p* < 0.0001) for ZNP^A^. Similarly, the cell viability was reported to be 67.14% (*p* < 0.0001) and 61.42% (*p* < 0.0001) at 50 and 100 μM concentrations of ZNP^B^. Recently, numerous studies have demonstrated its non-toxic nature [39,105,116,117,118]. In vivo examinations on blood, normal tissues, and inner organs, as well as in vitro testing on healthy and intact adult RBCs, found no adverse toxicity [106]. Furthermore, there is no indication of mutagenicity, carcinogenicity, or genotoxicity [119,120]. Moreover, a bio-availability study revealed minimum damage to organs following administration of very high doses of 500 mg/kg [121].

#### 3.3.2. Structural Changes in Cancer Cells

The morphology of the control appeared even, smooth, and flattened, maintaining their pristine condition when treated with ZNPs (Figure 6). Contrastingly, the treated A431 cells demonstrated characteristic apoptotic changes showing shrunken, spherical, and nonadherent cells with chromatin condensation. The influence of ZNPs on cancerous cells’ anatomy was shown to be dose- as well as size-dependent. Moreover, ZNP^A^ treatment was more effective, revealing a large collection of rounded, shriveled cell arrangement. Similar studies have demonstrated that after ZNP treatment, cells drastically change their morphology and clump together in the medium [122]. This is in agreement with earlier findings indicating finer nanomaterials exhibit higher cytotoxic action [123,124].

#### 3.3.3. MTT Assay

Cell growth inhibitory action of various ZNP samples at different concentrations displayed a dose-dependent effect (Figure 7). Only 91.46 ± 2.13, 57.23 ± 0.34, and 38.00 ± 1.27% (*p* < 0.0001) cells remained viable when treated with ZNP^A^ at a concentration of 5, 10, and 25 μM. However, ZNP^B^ was more cytotoxic to A431 cells, reducing cell viability to roughly 88.45 ± 1.13, 75.05 ± 0.92, and 55.68 ± 2.20% (*p* < 0.0001) at the same concentration range. Patel et al. [125] reported 52.71% viability of A431 cells at 25 µM concentration after 48 h exposure. The IC_50_ value of ZNP^A^ and ZNP^B^ was found to be ~9 μM and >25 μM, respectively. The literature survey reveals comparable IC_50_ estimates for ZNPs [36,126,127]. The reduction in cancer cell survival as per MTT data verifies ZNP-induced apoptosis in A431 cells [128].

#### 3.3.4. Effect of ZNPs on Intracellular ROS Generation

Following 24 h of incubation, ZNPs significantly boosted ROS formation in A431 cells, as evidenced by high cytosolic green fluorescence intensity (Figure 8a). The ROS generation by ZNPs depended on the amount of therapeutic dose, as previously reported [129,130]. Quantitative analysis of ROS revealed that 5 μM of ZNP^A^ increased ROS generation by 110.27% (non-significant), while ZNP^B^ at the same concentration enhanced ROS level by 107.54% (non-significant) (Figure 8b). Furthermore, ROS levels were elevated by 119.81% (*p* < 0.001) and 116.86% (*p* < 0.0001) compared to the control at 10 and 25 μM concentrations of ZNP^B^, respectively. Likewise, similar concentrations of ZNP^A^ led to the increase in ROS production by 133.81% (*p* < 0.001) and 129.36% (*p* < 0.01), respectively, over the control (Figure 8b).

#### 3.3.5. Effect of ZNPs on Chromatin Condensation

We further investigated chromatin condensation as an apoptotic feature using 4′, 6-diamidino-2-phenylindole (DAPI) labeling. A431 cells were incubated with ZNPs for 24 h prior to evaluation for DNA integrity. Compared to untreated cells, DAPI staining revealed significant chromatin condensation in treated cells compared to untreated control (Figure 9). Maximum chromatin condensation was observed at 25 μM concentration of ZNPs (Figure 9a,b). The shrunken nuclei in A431 cells are indicative of apoptotic cell death. ZNP^A^ and ZNP^B^ at 5 μM concentration induced generation of 11.14% (non-significant) and 10.38% (non-significant) apoptotic cells, respectively. At a concentration of 10 and 25 μM, 23.86% (*p* < 0.001) and 34.45% (*p* < 0.0001) apoptotic cells were observed for ZNP^A^, while approximately 18.49% (*p* < 0.01) and 29.25% (*p* < 0.001) of apoptotic cells were observed for ZNP^B^ at the same concentration (Figure 9b). These results allow the conclusion that the compounds promote apoptosis in A431 lung cancer cells.

The fundamental source of mitochondrial dysfunction and oxidative stress is believed to be intracellular ROS generation [131,132,133]. We found a progressive rise in intracellular ROS species upon increasing the concentration of ZNPs (Figure 8). In light of this, we hypothesize that ZNPs generate intracellular free radicals, which induces excessive oxidative stress in the treated cells. It has been demonstrated that the quality of ZnO nanocrystals diminishes with size, resulting in more interstitial zinc ions and oxygen vacancies [131,134] that eventually lead to the generation of ROS. This could explain the slightly better cytotoxic activity of ZNP^A^, which, due to its small size, generates more holes and vacancies, thus generating a higher amount of ROS.

Several theories have been put forth to explain how metallic NPs produce ROS [135,136]. In the literature, some speculate that ZNPs could generate a significant number of Zn^2+^ ions in solutions, especially in tissue culture mediums. This Zn^2+^ ion generation may act as one of the possible ROS inducers [137,138,139].

An alternative notion relating cytotoxic effects and apoptotic death is the existence of electron acceptor functional clusters on the NP exterior generated by crystal defects, as well as the disordered organization of electronic configurations typically occurring when a nanomaterial shrinks in size [131]. These ZNP structural aberrations may result in an enormous number of electron–hole pairs. This generates highly reactive free radicals, including the superoxide anion radical (O_2_^−•^) (from electrons) and the hydroxyl radical (from holes). The free radicals can migrate to the nanoparticle surface and can cause an increased formation of superoxide radicals, which leads to ROS accumulation and oxidative stress [140]. These radicals can oxidize and reduce macromolecules in the cellular milieu, causing serious oxidative damage to the cell.

Furthermore, substantial overexpression of the genes involved in ROS stress management has been reported [136]. Co-activation of these genes is abnormal for mitochondrial stress management and might be linked to defective ROS detoxification mechanisms in cells, potentially due to ZNP-mediated toxic effects [141,142]. Thus, ZNPs produce heterogeneous cytotoxicity harm to cells by excessively generating ROS [136]. Nonetheless, the precise fundamental physiological mechanism for ROS formation by ZNPs is still unresolved and needs to be thoroughly elucidated.

## 4. Conclusions

The aim of the present study was to evaluate the effect of extending the incubation period on the production of ZNPs. ZNPs with husk-like morphologies (ZNPs) were confirmed by SEM, while their wurtzite crystalline nature was confirmed by XRD. Additionally, UV-vis spectra showed a blue-shift in spectrum behavior. Our study concludes that smaller-sized particle, i.e., ZNP^A^, synthesized with a lower incubation period, had a more drastic effect on human epidermoid carcinoma cells than ZNP^B^, synthesized with an extended incubation time. ZNP^A^ samples showed increased nuclear condensation and increased ROS generation, which led to cell death and nuclear apoptosis. This research widens the possibilities for using ZNPs as chemotherapeutic agents.

## Data Availability

Not applicable.

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
