# Peer review of "Husk-like Zinc Oxide Nanoparticles Induce Apoptosis through ROS Generation in Epidermoid Carcinoma Cells: Effect of Incubation Period on Sol-Gel Synthesis and Anti-Cancerous Properties"

_biomedicines, 2023, doi:10.3390/biomedicines11020320_

Round 1

Reviewer 1 Report

In this work, Wardah et al. developed a zinc oxide nanoparticle (ZNP) for anti-cancer treatment. The husk-like (ZNP) is with diameter around 200 nm, showing ROS release to induce cytotoxicity in A431 cancer cells. This work is sounding. But some points must be explained or revised before consideration for publication.

1.     Is ZNP unstable? The UV-vis spectra in Figure 1 showed an increased absorption in 10 weeks. What’s the medium? I suggest the authors to investigate it in biological medium such as PBS and FBS.

2.     What’s the mechanism of ROS generation? ZNP is a type of inorganic photosensitizers. No external light sources used for ROS generation?

3.     The images in Figure 8A are blurred. Hard to distinguish nucleus of cancer cells. DAPI staining is recommended.

4.     Statistical difference analysis should be provided in Figure 5, Figure 7, Figure 8B, and Figure 9B.

5.     Whether ZNP would cause toxicity to normal tissue cells? This may discount their perspectives in medical applications.

Reviewer 2 Report

Alhoqail et al. present a comprehensive investigation documenting the method of synthesis, characteristics, and anti-cancer efficacy of husk-like zinc oxide nanoparticles (ZNPs). In their introduction the authors note the interest in ZNPs in a variety of industrial and cosmetic processes, noting their relatively harmless effects on people in terms of toxicity. They note that several different methods are available to create ZNPs, resulting in very different ultrastructural characteristics. The authors then posit that differing times of incubation could affect ZNP ultrastructure and then also posit that they could have anti-cancerous effects on human epidermoid carcinoma (HEC) cells as these cells would be the first contacted by ZNPs (presumably in a topical form).

It is standard to justify why an investigation is performed in the Introduction, with goals and hypotheses to be tested placed there. The reader has to wait until section 3.3 to learn about the anti-cancer literature involving ZNPs just prior to presenting supporting data. While section 3 is Results and Discussion, I would prefer that the authors provide at least a few statements that better justify the study in the Introduction.

The Methods section is appropriately detailed concerning how ZNPs were manufactured, how their physical and ultrastructural characteristics were assessed; further, the various biochemical methods, cell types and cell cultures were well detailed. The statistical analyses are noted, but the methods used are likely underpowered given that there are up to seven conditions with two groups with only n=3 per group/condition. I would like to know what preliminary information provided a statistical power of 0.8 or greater. Statistical power, or sensitivity, is the likelihood of a significance test detecting an effect when there actually is one. As the authors have set P< or = 0.05, I would like to know what power they have with that with their various analyses.

In Results, the two- and ten-week ZNPs had different physical characteristics and ultrastructural features. In the presentation of various parameters, either statistical significance is not noted at all in figures or legends, or legends mention significance without any seen in the figure. For example, figure 5 displays remarkable changes in hemolysis and cell viability, but there are no statistical remarks. In figure 7, 8 and 9, there are no statistical symbols, but the legend notes that there was significance. The fonts and styles of the figures are also variable, with the SD poorly visualized. Lastly, some sort of grading system concerning cell morphology with relevant features pointed out in the figures with arrows, etc., would have been helpful.

The Conclusion section is brief, and the conclusions drawn by the authors are purported to be supported by the preceding data.

In summary, while interesting, it is unclear if the authors have enough statistical power to have data that supports their conclusions with their limited number of experiments with numerous conditions. Also, the inattention to detail concerning the figures makes it difficult to know what is being conveyed in terms of either morphology or statistical significance.

Reviewer 3 Report

In the present manuscript, effect of incubation period on production of husk-like zinc oxide nanoparticles by a simple sol-gel method and their anti-cancerous properties have been determined. The nanoparticles show crystalline structure, ROS-generating abilities and cytotoxicity against skin cancer cell lines. Overall, the manuscript lacks certain evidences that can support their claims. My specific comments were provided below.

1.    In line number 211, unit of incubation time needs to be changed from min to weeks.
2.    In line number 250, authors have reported the morphological pattern of nanoparticles to be hexagonal which was observed with Scanning Electron Microscope. Evidence for this statement needs to be provided.
3.    In figure 8(B), the colour coding for different groups needs to be checked and corrected.
4.    In line number 427, there is a repeat in the term “following”.
5.    In figure 8(B), there is background fluorescence which is visible in the image.
6.    For supporting the hypothesis that cancer cells are undergoing apoptosis, flow cytometry can be a better option as the chromatin condensation is not a specific marker for apoptosis as chromatin condensation is seen in other cellular processes too.

Round 2

Reviewer 1 Report

I agree to publish it.

Author Response

We thank the learned referee for their positive feedback on our revised draft.

Reviewer 2 Report

The authors now include statistical significance, but they still do not address the matter of statistical power. They only have n=3 per condition with up to 7 different conditions presented. It does not matter what the P value is if the statistical power is inadequate. The authors must obtain assistance and must present us with the power present with their numbers. If the power is below 0.8 for the analyses presented, then they need to do more experiments.

Reviewer 3 Report

The authors have submitted the revised manuscript now. All the comments were satisfactorily addressed. The manuscript is now recommended for publication.

Author Response

We are highly thankful to the referee for positive remarks on our revised manuscript. 

Round 3

Reviewer 2 Report

No further comments.